# Transient Decrease in Incidence Rate of Maternal Primary Cytomegalovirus Infection during the COVID-19 Pandemic in Japan

**DOI:** 10.3390/v15051096

**Published:** 2023-04-29

**Authors:** Kuniaki Toriyabe, Asa Kitamura, Miki Hagimoto-Akasaka, Makoto Ikejiri, Shigeru Suga, Eiji Kondo, Masamichi Kihira, Fumihiro Morikawa, Tomoaki Ikeda

**Affiliations:** 1Department of Obstetrics and Gynecology, Mie University Graduate School of Medicine, Tsu 514-8507, Japan; 2Department of Obstetrics and Gynecology, National Hospital Organization Mie Chuo Medical Center, Tsu 514-1101, Japan; 3Department of Clinical Laboratory, Mie University Hospital, Tsu 514-8507, Japan; 4Institute for Clinical Research, National Hospital Organization Mie National Hospital, Tsu 514-0125, Japan; 5Mie Association of Obstetricians and Gynecologists, Tsu 514-0003, Japan

**Keywords:** primary cytomegalovirus infection, pregnancy, CMV screening, congenital cytomegalovirus infection, immunoglobulin G, seroconversion, SARS-CoV-2, COVID-19, pandemic

## Abstract

This study evaluated the impact of the coronavirus disease 2019 (COVID-19) pandemic on the occurrence of maternal primary cytomegalovirus (CMV) infection in Japan. We performed a nested case-control study using data from maternal CMV antibody screening under the Cytomegalovirus in Mother and infant-engaged Virus serology (CMieV) program in Mie, Japan. Pregnant women with negative IgG antibodies at ≤20 weeks of gestation who were retested at ≥28 weeks were enrolled. The study period was divided into 2015–2019 as the pre-pandemic and 2020–2022 as the pandemic period, and the study site included 26 institutions conducting the CMieV program. The incidence rate of maternal IgG seroconversion was compared between the pre-pandemic (7008 women enrolled) and pandemic (2020, 1283 women enrolled; 2021, 1100 women; and 2022, 398 women) periods. Sixty-one women in the pre-pandemic period and five, four, and five women during 2020, 2021, and 2022, respectively, showed IgG seroconversion. The incidence rates in 2020 and 2021 were lower (*p* < 0.05) than that in the pre-pandemic period. Our data suggest a transient decrease in the incidence of maternal primary CMV infection in Japan during the COVID-19 pandemic, which could be due to prevention and hygiene measures taken at the population level.

## 1. Introduction

Cytomegalovirus (CMV) is a pathogen associated with mother-to-child transmission and is known to cause damage to the fetal central nervous system. Besides neurological dysfunctions, hepatological or hematological dysfunctions are well-known damage in infants. CMV infections are established through human body fluids, such as urine and saliva, from others. Maternal primary CMV infection during pregnancy often occurs through young children’s urine or saliva [1].

Both maternal CMV infection before conception (non-primary CMV infection) and after conception (primary CMV infection) can cause congenital CMV (cCMV) transmission. Serological screening, including assessment of CMV immunoglobulin (Ig) G and IgM antibody levels and IgG antibody avidity, is used to identify mothers with primary CMV infection, which is associated with a higher risk of cCMV transmission than that of non-primary CMV infection. Maternal primary CMV infection is confirmed with CMV IgG antibody seroconversion during pregnancy and is suspected by a set of positive CMV IgG antibodies, positive IgM antibodies, and low IgG antibody avidity [1].

Maternal CMV antibody screening is performed in Mie, Japan under a program called the Cytomegalovirus in Mother and infant-engaged Virus serology (CMieV) program [2,3,4,5]. The program is conducted to identify mothers with primary CMV infection and subsequent cCMV infection in their infants. In the program, pregnant women with CMV IgG antibody seroconversion during pregnancy are picked up, and their infants are tested for cCMV infection.

In Wuhan, China, coronavirus disease 2019 (COVID-19) emerged at the end of 2019. COVID-19 spread rapidly across China and became a worldwide pandemic [6,7]. In Japan, the first wave of COVID-19 came in 2020. On 7 April 2020, the first state of emergency was declared by Japan government. In response to that, a prefectural state of emergency was declared in Mie, Japan on 10 April 2020. After that, additional three states of emergency were declared in Japan. The last (fourth) state was released on 30 September 2021. The COVID-19 vaccination began in April 2021. COVID-19 is caused by severe acute respiratory syndrome coronavirus 2 (SARS-CoV-2). This disease has diverse sequelae in patients of all age groups [8,9]. Respiratory droplets and close personal contact are the major sources of SARS-CoV-2 transmission through breathing, sneezing, coughing, and normal speech [8,9,10]. Since the beginning of the SARS-CoV-2 spread and the COVID-19 pandemic in 2020, considerable behavioral changes in persons, communities, and societies have been observed all over the world, including Japan. Personal hygiene measures, community closures (such as closures of public schools and workplaces), and social distancing were implemented in many countries during the COVID-19 pandemic.

After the onset of the COVID-19 pandemic, the epidemiology of some viral infections, such as influenza virus and respiratory syncytial virus, has changed in several countries. Under the situation, decreases in the incidence of infant cCMV infections have been reported in Portugal in 2020, Spain in 2020–2021, and the USA in 2020–2021 [11,12,13]. The actual situation of infant cCMV infection or maternal primary CMV infection in Japan is not known.

To our knowledge, to date, there is no study that reported a decrease in maternal primary CMV infection during the COVID-19 pandemic. Our aim was to evaluate the impact of the COVID-19 pandemic on the occurrence of maternal primary CMV infection during pregnancy in Japan using the CMieV cohort that was continued across the COVID-19 pre-pandemic and pandemic periods.

## 2. Methods

This was a nested case-control study using data from the CMieV program. The program has contained a maternal CMV antibody screening and a newborn urine CMV DNA test and has been a population-based, mother–child, observational, prospective cohort study.

### 2.1. Enrollment of Pregnant Women from the CMieV Program

As per the CMieV program, pregnant women are tested for serum CMV IgG plus IgM antibodies at ≤20 weeks of gestation. Pregnant women with negative CMV IgG antibody results are educated on how to prevent CMV transmission from other persons and instructed to avoid direct contact with infectious body fluids, such as urine and saliva of children and semen of male partners, during pregnancy. CMV IgG antibody testing is repeated at ≥28 weeks of gestation. Pregnant women with CMV IgG antibody seroconversion (negative results at ≤20 weeks and positive results at ≥28 weeks of gestation) are considered as having primary CMV infection during pregnancy. Their neonates are tested for the presence of CMV DNA in fresh urine specimens obtained within one week after birth. Neonates with positive fresh urine CMV DNA results are diagnosed with cCMV infection. Using the CMV DNA-positive fresh urine, viral isolation tests are performed [2,3].

In the current study, pregnant women who were CMV IgG negative at ≤20 weeks of pregnancy and were retested for CMV IgG antibodies at ≥28 weeks were retrospectively enrolled in the CMieV program. The study period was from January 2015 to June 2022 (January 2015–December 2019 as the COVID-19 pre-pandemic period and January 2020–June 2022 as the COVID-19 pandemic period). The study site included 26 CMieV program institutions in Mie, Japan (1 university hospital, 3 public hospitals, 4 private hospitals, and 18 private clinics; Mie University Hospital, Municipal Yokkaichi Hospital, National Hospital Organization Mie Chuo Medical Center, Saiseikai Matsusaka General Hospital, Mitaki General Hospital, Yonaha Okanoue Hospital, Shiroko Women’s Hospital, Morikawa Hospital, Yonaha Ladies’ Clinic, Kawai Obstetrics and Gynecology, Suzuki Ladies’ Clinic, Takeda Obstetrics and Gynecology, Obata Ladies’ Clinic, Midorigaoka Clinic, St. Rose Clinic, Yanase Clinic, Minami Obstetrics and Gynecology, Tamaishi Obstetrics and Gynecology, Terada Obstetrics and Gynecology, Mie Ladies’ Clinic, Kanamaru Obstetrics and Gynecology, Jiho Obstetrics and Gynecology, Tsunishi Obstetrics and Gynecology, Nishiyama Obstetrics and Gynecology, Fujimoto Clinic, and Miyazaki Obstetrics and Gynecology).

### 2.2. CMV IgG Antibody and CMV DNA Tests

CMV IgG antibody test was performed using enzyme-linked immunosorbent assay (ELISA) kits (Denka Co., Ltd., Tokyo, Japan). In the current study, negative and positive IgG antibody results were defined by IgG titers of ≤3.9 and ≥4.0 ELISA values, respectively.

Real-time polymerase chain reaction (PCR) was performed for CMV DNA detection in neonatal fresh urine samples as previously described [2]. Gene name was UL123, and gene description was regulatory protein IE1. Real-time PCR was performed using an ABI 7900 Fast Real-time PCR system (Applied Biosystems, Foster City, CA, USA). A mixture consisted of the QuantiTect Multiplex PCR Kit (Qiagen, Hilden, Germany), sense (5′-TTTTTAGCACGGGCCTTAGC-3′) and antisense (5′-AAGGAGCTGCATGATGTGACC-3′) primers, TaqMan Probe (5′-FAM-TGCAGTGCACCCCCCAACTTGTT-TAMRA-3′) (Applied Biosystems), and the sample DNA. Viral DNA was extracted from neonatal fresh urine samples using a QIAamp DNA Blood Mini Kit (Qiagen). DNA was amplified with pre-cycling holds. As the annealing steps and the extension steps were performed at the same temperature, these steps were simultaneously performed. Samples were analyzed in duplicate using the AcroMetrix CMVtc Panel (Applied Biosystems) to generate the standard curve. The CMV DNA test was performed at the Mie University Hospital (Mie, Japan), and positive CMV DNA results were defined by ≥200 copies/mL of CMV DNA. In viral isolation tests, the presence of cytopathic effect was determined using human fetal fibroblasts up to 6 weeks after culture onset. The culture medium was replaced with a fresh one every week. Viral isolation tests were performed at National Hospital Organization Mie National Hospital (Mie, Japan).

### 2.3. Comparison of the Incidence Rates of Maternal Primary CMV Infection and Neonatal cCMV Infection between the COVID-19 Pre-Pandemic (2015–2019) and Pandemic (2020–2022) Periods

The incidence rate of maternal CMV IgG antibody seroconversion was compared between the COVID-19 pre-pandemic (2015–2019) and pandemic (2020, 2021 and 2022) periods. The whole of the years 2020 and 2021 (January–December) were included, whereas in 2022 only the first 6 months (January–June) were included. Both annual (January–December) and semi-annual (January–June and July–December) incidence rates were studied. Similarly, the incidence of positive fresh urine CMV DNA results in neonates was compared between the COVID-19 pre-pandemic and pandemic periods. The annual (January–December) incidence rates were also determined. The incidence rate of maternal CMV IgG antibody seroconversion in each age group (teens, twenties, and thirties and over) and each parity group (para 0, 1, and ≥2) was compared between the COVID-19 pre-pandemic and pandemic periods. Fisher’s exact test was used for comparisons using SPSS software (ver. 27; IBM Corp., Armonk, NY, USA), and a *p* value of ≤0.05 was considered statistically significant.

## 3. Results

The numbers and clinical characteristics (age and parity; however, no information in terms of employment) of pregnant women enrolled during the COVID-19 pre-pandemic (2015–2019) and pandemic (2020, 2021, and 2022) periods are shown in Table 1. A total of 7008 pregnant women were enrolled during the pre-pandemic period, and 1283, 1100, and 398 pregnant women were enrolled during 2020 (January–December), 2021 (January–December), and 2022 (January–June only), respectively. The median age was 30 years through all the COVID-19 pre-pandemic and pandemic periods. Median parity was para 0 during the COVID-19 pre-pandemic period, and para 1 through all the pandemic periods. Sixty-one (0.9%) pregnant women during the pre-pandemic period and five (0.4%), four (0.4%), and five (1.3%) pregnant women during 2020, 2021, and 2022 (the pandemic period), respectively, showed CMV IgG antibody seroconversion during pregnancy. Eighteen (0.3%) pregnant women in the pre-pandemic period, and two (0.2%), none (0.0%), and one (0.3%) pregnant woman in 2020, 2021, and 2022, respectively, had neonates with positive fresh urine CMV DNA results in real-time PCR tests and positive cytopathic effect results in viral isolation tests.

The incidence rates of maternal CMV IgG antibody seroconversion during pregnancy in 2020 (*p* = 0.0451) and 2021 (*p* = 0.0486) were significantly lower, and that in 2022 (*p* = 0.2805) was not significantly different from that in the pre-pandemic period (Figure 1). The incidence rates of neonatal fresh urine CMV DNA-positive results in 2020, 2021, and 2022 were not significantly different from that in the pre-pandemic period (Figure 1).

The annual (January–December) and semi-annual (January–June and July–December) incidence rates of maternal CMV IgG antibody seroconversion during 2015–2022 are shown in Figure 2. The annual incidence rate of 0.4% in 2020 and 2021 was the lowest during the study period (2015–2022). The semi-annual incidence of 0.0% in the second half (July–December) of 2020 was the lowest, whereas that of 0.2% in the first half (January–June) of 2021 was the second lowest during 2015–2022. The annual and semi-annual incidence rates of neonatal fresh urine CMV DNA-positive results during 2015–2022 are shown in Figure 3. An annual incidence rate of 0.0% was observed in 2021. Moreover, a semi-annual incidence rate of 0.0% was observed in the three consecutive periods (the second half of 2020 and both halves of 2021).

The incidence rates of maternal CMV IgG antibody seroconversion during pregnancy in each age group (teens, twenties, and thirties and over) and each parity group (para 0, para 1, and para ≥ 2) during the COVID-19 pre-pandemic and the COVID-19 pandemic periods are shown in Figure 4 (Upper). There was no maternal CMV IgG antibody seroconversion case in the teen group during the COVID-19 pandemic period. During the COVID-19 pandemic period, the incidence rate of maternal CMV IgG antibody seroconversion was lower in all age groups and all parity groups than that in the COVID-19 pre-pandemic period. Significant differences were found only in the group of thirties and over (*p* = 0.0480) and in the group of para 0 (*p* = 0.0492) for age and parity, respectively. Monthly incidences of maternal CMV IgG antibody seroconversion during the COVID-19 pandemic period are shown in Figure 4 (Lower). From April 2020 to September 2021, the period during which a total of four states of emergency were declared in Japan, monthly incidences of maternal CMV IgG antibody seroconversion decreased.

## 4. Discussion

In the present study, the incidence rate of maternal CMV IgG antibody seroconversion in 2020–2021 was significantly lower, and that in 2022 was not significantly different from that in the COVID-19 pre-pandemic period. This indicates that the incidence of maternal primary CMV infection transiently decreased during the COVID-19 pandemic in Mie, Japan; however, this decrease was reversed in 2022 to the same level as the COVID-19 pre-pandemic period. Further, the semi-annual incidence rates in the second half of 2020 and the first half of 2021 were low, whereas those in the second half of 2021 and the first half of 2022 were as high as in the COVID-19 pre-pandemic period.

In the present study, significant differences in the incidence rate of maternal CMV IgG antibody seroconversion were found only in the group of thirties and over and in the group of para 0 for age and parity, respectively. We did not interpret the present results as indicating that only the thirties and over group decreased in age groups and only the para 0 group decreased in parity groups. We speculated that the small sample sizes of the teens and twenties groups and the para 1 and para 2 groups might have prevented us from observing a significant decrease in the incidence rate of maternal CMV IgG antibody seroconversion. First, the thirties and over group had a higher proportion of pluripara than the other age groups, making it difficult to link the thirties and over and para 0. Second, the thirties and over group actually had the largest sample size by age group, and the para 0 group also has the largest sample size by parity group. Thus, we assumed that the incidence rates of maternal CMV IgG antibody seroconversion in all age groups and all parity groups decreased during the COVID-19 pandemic period compared to that in the pre-pandemic period.

We reported in the previous literature that the incidence rate of maternal CMV IgG antibody seroconversion was significantly (*p* < 0.001) higher in teens (5.0%) than in older pregnant women (twenties, 0.8%; thirties and over, 0.6%) [14]. As the teenage pregnant women were all para 0, difference in the route of CMV transmission during pregnancy was considered to be one factor of the high incidence rate of primary CMV infection in them. Teenage pregnant women were estimated to have been infected with CMV through body fluids other than their own young children, i.e., semen of sexual partner, while older and parous pregnant women were estimated to have been infected mainly through their own young children’s body fluid [14]. Although the incidence rate of maternal CMV IgG antibody seroconversion in teenage pregnant women was high compared to older pregnant women in the current study, the number of teenagers was very small compared to that of older pregnant women. Though the decrease in the incidence rate of maternal CMV IgG antibody seroconversion during the COVID-19 pandemic period in this study appeared to be greater among teenagers than among the other age groups, the size of the teenager age group was considerably smaller than that of other age groups. Therefore, we conclude that the incidence rate of maternal primary CMV infection did not decrease specifically among teenagers but rather decreased among all age groups in Mie, Japan during the COVID-19 pandemic period.

In the CMieV cohort, the number of mothers with primary CMV infection significantly decreased during 2020–2021 compared to that in the COVID-19 pre-pandemic period. These findings can be explained as follows: First, extensive changes in personal and social behavior and the hygiene habits in daily life at the population level have reduced the transmission of SARS-CoV-2 and resulted in the end of the COVID-19 pandemic. One of the major routes of CMV transmission is direct contact with other’s body fluids, including urine and saliva. In Japan, the first, second, and third waves of COVID-19 came in 2020. The fourth and fifth wave came in 2021, and the sixth wave came in 2022. On 7 April 2020, a state of emergency was declared by Japan’s government (the first state, 7 April 2020–25 May 2020; the second state, 8 January 2021–21 March 2021; the third state, 25 April 2021–20 June 2021; the fourth state, 12 July 2021–30 September 2021). In response to the first state, a prefectural state of emergency was declared in Mie, Japan on 10 April 2020. COVID-19 vaccination began in April 2021. Under the state of emergency, residents were strongly urged to refrain from leaving their houses and moving unnecessarily, except when necessary to maintain their daily lives. Residents were strongly requested to take thorough actions to avoid enclosed, dense, and closed areas in cases when leaving the house was unavoidable. Since the major route of familial transmission of CMV to mothers is from young boys and girls attending daycare centers, this could be one of the factors contributing to the decline in maternal CMV infection during the pandemic [15,16]. Recent studies have reinforced the role of young children in the epidemiology of CMV infections [17]. Behavioral measures in daily lives incorporating reduced direct contact with body fluids from young children decrease CMV antibody seroconversion in mothers during pregnancy [18].

Universal mask use and increased frequency of hand washing in daily lives during the pandemic might have also contributed to the decreased maternal primary CMV infection during the pandemic compared to that of the COVID-19 pre-pandemic period, as demonstrated in studies on other viral infections, such as influenza virus and respiratory syncytial virus [19,20,21]. Villaverde et al. reported that the necessity to keep mothers and young children at home during the COVID-19 pandemic period might have contributed to the reduction in primary CMV transmission in mothers during pregnancy. As there has been little progress in preventing neonatal cCMV infection, these findings provide new data supporting the education of mothers during pregnancy by obstetric practitioners on steps to decrease the risk of domestic (young child to mother) CMV infection [12]. Fernandez et al. demonstrated a decrease in the occurrence of neonatal cCMV infection comparing 2019 and 2020 through neonatal saliva cCMV screening in Portugal [11]. The findings by Fernandez et al. further verify that the current study hypothesis is correct.

In the current study, we demonstrated a transient decrease in the incidence rate of maternal primary CMV infection during pregnancy in Mie, Japan during the COVID-19 pandemic period (2020 and 2021) compared to the COVID-19 pre-pandemic period (2015–2019); however, we could not statistically show a decrease in the incidence rate of neonatal cCMV infection in the same population as maternal primary CMV infection. This might be due to a statistically small number of neonatal cCMV infection observations compared to the three previous studies performed not in Japan [11,12,13]. Although a decrease in neonatal cCMV infection under the COVID-19 pandemic could not be statistically proven in the current study, it is possible that neonatal cCMV infection rate along with maternal primary CMV infection also decreased in 2020 and 2021 during the COVID-19 pandemic in Japan.

In conclusion, the findings of the current study suggest a significant and transient decrease in the incidence of maternal primary CMV infection during the COVID-19 pandemic period in Japan compared to the COVID-19 pre-pandemic period. This was probably due to the prevention of SARS-CoV-2 transmission through personal and social hygiene measures taken at the population level to end the COVID-19 pandemic.

## Figures and Tables

**Figure 1 viruses-15-01096-f001:**
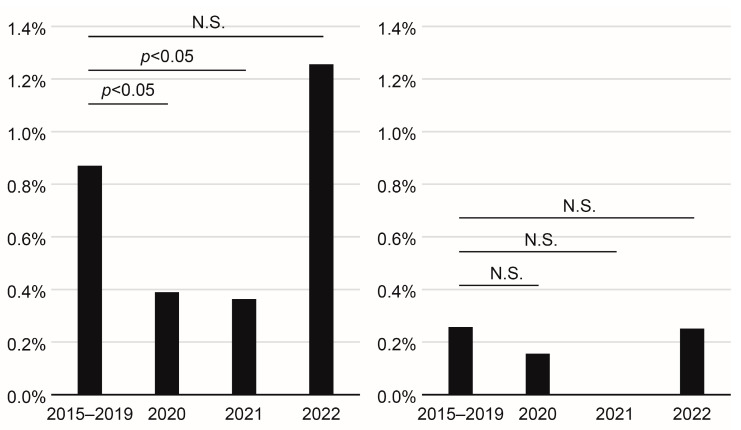
Comparison of the incidence rates of maternal cytomegalovirus (CMV) immunoglobulin G antibody seroconversion during pregnancy (**left**) and neonatal fresh urine CMV DNA-positive results (**right**) between the coronavirus disease 2019 pre-pandemic (2015–2019) and the pandemic (2020, 2021, and 2022) periods. N.S., not significant.

**Figure 2 viruses-15-01096-f002:**
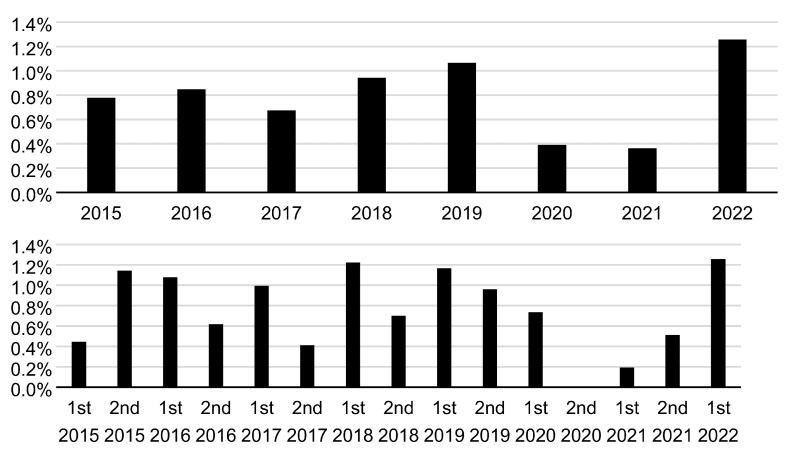
Annual (**upper**) and semi-annual (January–June and July–December) (**lower**) incidence rates of maternal cytomegalovirus immunoglobulin G antibody seroconversion during pregnancy through the study period (2015–2022).

**Figure 3 viruses-15-01096-f003:**
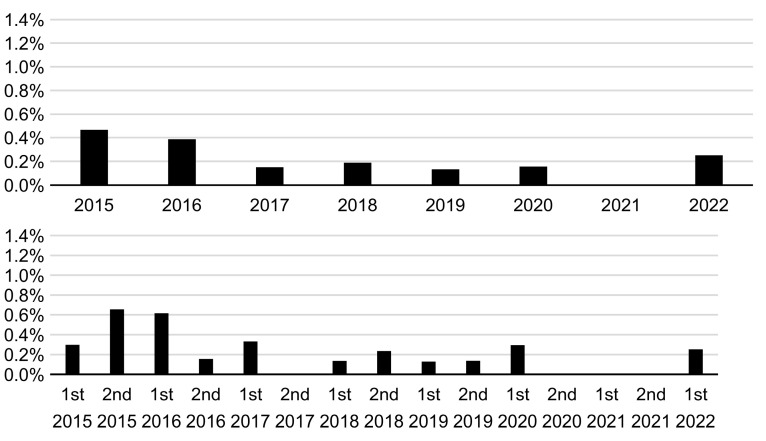
Annual (**upper**) and semi-annual (January–June and July–December) (**lower**) incidence rates of neonatal fresh urine cytomegalovirus DNA-positive results throughout the study period (2015–2022).

**Figure 4 viruses-15-01096-f004:**
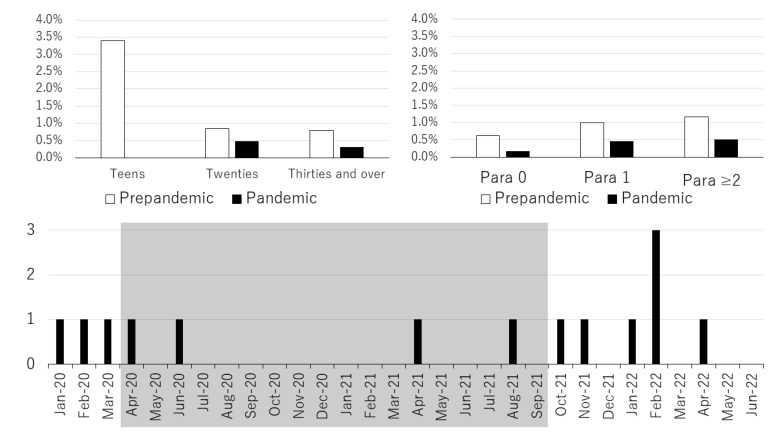
Incidence rate of maternal cytomegalovirus immunoglobulin G antibody seroconversion in each age group (**upper-left**) and each parity group (**upper-right**) during the coronavirus disease 2019 pre-pandemic (2015–2019) and pandemic (2020–2022) periods. There were no cases in the teen group during the COVID-19 pandemic period. Significant differences were found only in the thirties and over group (**upper-left**) and in the para 0 group (**upper-right**). Monthly incidences of maternal cytomegalovirus immunoglobulin G antibody seroconversion during the COVID-19 pandemic period (**lower**). The shaded area indicates the period during which a total of four states of emergency were declared in Japan.

**Table 1 viruses-15-01096-t001:** Characteristics of study participants according to year of enrollment (pre-pandemic, 2015–2019; pandemic, 2020, 2021, and 2022).

Pre-Pandemic or Pandemic Period of the COVID-19	Pre-Pandemic		Pandemic	
Year	2015–2019	2020	2021	2022 *
Enrolled pregnant women, *n*	7008	1283	1100	398
Age (years), median: range	30: 16–47	30: 16–45	30: 17–45	30: 19–44
Age groups				
Teens	2.1%	2.0%	1.5%	0.2%
Twenties	47.1%	45.1%	43.7%	43.8%
Thirties and over	50.8%	52.9%	54.8%	55.9%
Parity (para), median: range	0: 0–5	1: 0–3	1: 0–5	1: 0–6
Parity groups				
Para 0	50.3%	46.3%	47.3%	44.6%
Para 1	37.4%	38.6%	34.9%	38.0%
Para ≥ 2	12.3%	15.1%	17.8%	17.4%

COVID-19, coronavirus disease 2019. * January–June only.

## Data Availability

The datasets used and/or analyzed during this study are available from the corresponding author upon reasonable request.

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
