# Peer review of "Transient Decrease in Incidence Rate of Maternal Primary Cytomegalovirus Infection during the COVID-19 Pandemic in Japan"

_viruses, 2023, doi:10.3390/v15051096_

Round 1
Reviewer 1 Report
This study uses archival information collected in a unique program in Japan (CMVieV) that tracks cytomegalovirus seroconversion of pregnant women at 26 institutions. The authors compared seroconversion rates in pregnant women prior to the pandemic with those in pregnant women during the height of the pandemic (2020 and 2021) and during the first six months of 2022. The authors found that seroconversion in pregnant women during the height of the pandemic was lower than in women pre-pandemic. However, the incidence of congenital CMV infection was not different across these time periods, likely because of the small numbers of infants assessed in this study group.
Comments: This is a nicely written succinct manuscript that captures the interesting decrease in seroconversion rates in pregnant women during the pandemic. The authors propose that this is likely because of the measures taken during the pandemic to drastically limit person to person contact outside of family units and also the heightened awareness of handwashing and use of masks.
1. Is there any information in the database in terms of the employment of the women who did seroconvert? E.g. Presumably those employed in hospitals or clinics or essential services etc may have continued to work throughout the pandemic and were at higher risk for contact than those who remained at home the entire time of the pandemic. This information might help to support the proposal the authors are making that the reason for the reduced seroconversion was because most women stayed at home during the height of the pandemic.
2. The authors have performed statistics in Figure 1. Is it possible to also perform statistics on Fig 2 or Figure 4 to determine if the lower values were significant?
Reviewer 2 Report
The finding is not unexpected but it is interesting. It is also too long. It would make a great Brief Communication with just Table 1 and Fig 4.
Methodological issues....
Negative and positive antibody titres were defined as <3.9 and >4 but no units are given. Is this a log scale?
“Gene symbol” is the wrong word…and gene names should be in italics
6 weeks is a very long time to assess CPE in a single culture. Did uninfected cells remain healthy? Were cultures split and re-established in new medium? Are these data presented?
Table 1 does not need the “parity groups” and should include the data presented in Figs 1-3….which are then not required
The extent of the covid lockdown is not provided until the Discussion and then is hard to follow as written. Why not do a line graph of seroconversion each month with shaded blocks to represent lockdown. This could be fig 4C.
